# Untargeted lipidomics analysis in women with morbid obesity and type 2 diabetes mellitus: A comprehensive study

**Laia Bertran**[1], **Jordi Capellades**[2], **Sonia Abelló**[3], **Carmen Aguilar**[1], **Teresa Auguet**[1], **Cristóbal Richart**[1] *

1 Department of Medicine and Surgery, Study Group on Metabolic Diseases Associated with Insulin-Resistance (GEMMAIR), Rovira i Virgili University, Hospital Universitari de Tarragona Joan XXIII, IISPV, Tarragona, Spain, 2 Department of Electronic, Electric and Automatic Engineering, Higher Technical School of Engineering, Rovira i Virgili University, IISPV, Tarragona, Spain, 3 Scientific and Technical Service, Rovira i Virgili University, Tarragona, Spain

* crichartjurado@gmail.com

**Data Availability Statement:** All relevant data are within the manuscript and its Supporting Information files.

## Abstract

There is a phenotype of obese individuals termed metabolically healthy obese that present a reduced cardiometabolic risk. This phenotype offers a valuable model for investigating the mechanisms connecting obesity and metabolic alterations such as Type 2 Diabetes Mellitus (T2DM). Previously, in an untargeted metabolomics analysis in a cohort of morbidly obese women, we observed a different lipid metabolite pattern between metabolically healthy morbid obese individuals and those with associated T2DM. To validate these findings, we have performed a complementary study of lipidomics. In this study, we assessed a liquid chromatography coupled to a mass spectrometer untargeted lipidomic analysis on serum samples from 209 women, 73 normal-weight women (control group) and 136 morbid obese women. From those, 65 metabolically healthy morbid obese and 71 with associated T2DM. In this work, we find elevated levels of ceramides, sphingomyelins, diacyl and triacylglycerols, fatty acids, and phosphoethanolamines in morbid obese vs normal weight. Conversely, decreased levels of acylcarnitines, bile acids, lyso-phosphatidylcholines, phosphatidylcholines (PC), phosphatidylinositols, and phosphoethanolamine PE (O-38:4) were noted. Furthermore, comparing morbid obese women with T2DM vs metabolically healthy MO, a distinct lipid profile emerged, featuring increased levels of metabolites: deoxycholic acid, diacylglycerol DG (36:2), triacylglycerols, phosphatidylcholines, phosphoethanolamines, phosphatidylinositols, and lyso-phosphatidylinositol LPI (16:0). To conclude, analysing both comparatives, we observed decreased levels of deoxycholic acid, PC (34:3), and PE (O-38:4) in morbid obese women vs normal-weight. Conversely, we found elevated levels of these lipids in morbid obese women with T2DM vs metabolically healthy MO. These profiles of metabolites could be explored for the research as potential markers of metabolic risk of T2DM in morbid obese women.

**Funding:** The author(s) received no specific funding for this work.

**Competing interests:** The authors have declared that no competing interests exist.

## Introduction

Obesity has become as a public health issue over the world during the past few decades [1]. The prevalence of obesity has increased drastically during the last years, being linked to a higher mortality rate than underweight individuals [2]. The excessive accumulation of body fat induces some metabolic abnormalities, such as cardiovascular diseases, metabolic syndrome, Type 2 Diabetes Mellitus (T2DM), dyslipidemia, hypertension, metabolic dysfunction-associated steatotic liver disease, among others [3,4].

T2DM used to be initiated by systemic insulin resistance, along with a disruption in pancreatic β-cell insulin secretion [5]. Given that the incidence of T2DM is often increased linked to that of obesity, both conditions were recognized as epidemic by the World Health Organization [6].

There is a proportion of obese people with a much-decreased risk of cardiometabolic abnormalities, which commonly known as metabolically healthy obese subjects [7]. Despite the absence of a stated definition, normal glucose and lipid metabolism measures, as well as the absence of hypertension, are frequently used as diagnostic indicators [8]. The metabolically healthy obese phenotype seems to represent a transitional stage between the healthy individual and the disease, thus providing a model for studying the mechanisms linking obesity and T2DM [9].

Morbid obesity is a chronic condition for which no effective non-surgical treatment other than bariatric surgery has been found [10]. Hence, assessing those patients with Metabolically Healthy morbid obese versus those morbid obese patients with a major detrimental condition such as T2DM, may help us to identify the determining factors of this transient condition and better understand obesity and T2DM pathophysiology [11,12].

Omics-based technologies (genomics, transcriptomics, proteomics and metabolomics) were suggested to provide an advanced understanding of obesity etiology and its comorbidities [13]. A subfield of metabolomics called lipidomics focuses on measuring the different lipid species that are present in a particular biological system [14]. The great chemical diversity of lipids and their dynamic changes in response to physiological and environmental factors makes them interesting for their determination and quantification in the study of certain biological processes [15]. Recent findings allow us to identify by lipidomics many plasma lipids as mediators of metabolic dysfunction [16]. Some lipidomics studies on obesity and T2DM reported characteristic lipid profiles in these patients [17–21]. However, many of these studies have small or heterogeneous cohorts in terms of age, sex and degree of obesity.

In a previous study, we performed an untargeted metabolomics analysis in a homogeneous cohort of adult women with morbid obese compared to normal-weight women. In this work, we found some characteristic metabolite levels, mainly lipids, that allow us to differentiate the metabolome between morbid obese women who are metabolically healthy and those with associated T2DM [22]. In this sense, since it was a preliminary study, now we want to validate those findings assessing an untargeted lipidomics analysis in serum samples of the same well-characterized cohort of women with morbid obese to define the characteristic lipid profile in metabolically healthy morbid obese subjects and in morbid obese patients with associated T2DM.

## Material and methods

### Participants

In this study, we performed a liquid chromatography coupled to a mass spectrometer (LC/MS)-based untargeted lipidomics analysis in serum samples from 209 women. These women

were first classified depending on they were normal-weight (body mass index (BMI) 19–25 kg/m$^2$, n = 73) as the control group, or with morbid obese (BMI >40 kg/m$^2$, n = 136). Morbid obese group of subjects were scheduled to undergo a laparoscopic bariatric surgery. The normal-weight cohort was composed of volunteer women who met the criteria for metabolic health. The study cohort of this study was composed by only women to evaluate a homogenous cohort of subjects to avoid the interference of confounding factors such as sex. In this regard, it was previously reported that men and women differ substantially regarding body composition, energy imbalance, and hormones [23,24]. In addition, previous studies have informed about sex-specific differences in lipid and glucose metabolism [25].

The study was approved by the institutional review board (Institut Investigació Sanitària Pere Virgili CEIm; 23c/2015), and all participants gave written informed consent. This study was conducted retrospectively by accessing the patient data for the first time on June 5, 2023 and for the last time on January 18, 2024. During the data collection and the data analysis, the authors did not have access to the identification of the patients with the case studies because they worked on a blind and encrypted data base.

Patients who had an acute illness, acute or chronic inflammatory or infective diseases, or end-stage malignant disease were excluded from this study. Menopausal women and women receiving contraceptive treatment were also excluded. In addition, women with an alcohol intake exceeding 10g per day and recurrent smokers have been excluded from this study.

## Subclassification of the morbid obese cohort

Considering the 136 women with morbid obesity, 71 were previously diagnosed with T2DM meeting the diagnostic criteria of the American Diabetes Association [26,27]; meanwhile the other 65 were metabolically healthy morbid obese. Patients with metabolic syndrome without T2DM or pre-diabetes were initially excluded from this study due to the lack of homogeneity inside the group.

## Anthropometrical and biochemical variables

Anthropometrical evaluation included the age, the measurement of weight and height to calculate the BMI, and the waist–hip ratio. Blood samples were extracted by specialized nurses through a BD Vacutainer® system after overnight fasting, just before bariatric surgery in the case of the morbid obese cohort. Blood samples were obtained in ethylenediaminetetraacetic acid tubes, which were separated into plasma and serum aliquots by centrifugation (3500 rpm, 4˚C, 15 min), and then stored at −80˚C until processing. Biochemical analyses included glucose, insulin, glycated haemoglobin A1c (HbA1c), total cholesterol, high density lipoprotein-cholesterol (HDL-C), low density lipoprotein-cholesterol (LDL-C), and triglyceride levels, which were performed through a conventional automated analyzer.

## Samples preparation for the LC/MS analysis

Serum metabolites were extracted in 12μL of isopropanol and vortex-mixing for 10 seconds, incubated at 4˚C for 30 min and centrifuged (at 12000 rpm for 10 min at 4˚C).

## LC/MS setup

LC/MS was performed with a Thermo Scientific Vanquish Horizon Ultra High Performance LC system interfaced with a Thermo Scientific Orbitrap ID-X Tribrid Mass Spectrometer (Thermo Scientific, Waltham, MA, USA).

Metabolites were separated by reverse-phase chromatography with an Acquity Ultra Performance LC C18-RP (ACQUITY UPLC BEH C18 1.7μM, Waters). Mobile phase A was acetonitrile/water (60:40) (10mM ammonium formate), and mobile phase B was isopropanol/acetonitrile (90:10) (10mM ammonium formate). Solvent modifiers were used to enhance ionization and to improve the LC resolution in positive and negative ionization mode. Separation was conducted under the following gradient: 0–2 min, 15–30% B; 2–2.5 min, 48% B; 2.5–11 min, 82% B; 11–11.5 min, 99% B; 11.5–12 min, 99% B; 12–12.1 min, 15% B; 12.1–15 min, 15% B.)

For MS detection, heated electrospray ionization settings were set in positive and negative ionization modes as follows: source voltage, 3.5 kV (positive), 2.8 kV (negative); ion transfer tube, 300˚C; vaporizer temperature, 300˚C; sheath gas (N2) flow rate, 50 a.u.; auxiliary gas (N2) flow rate, 10 a.u.; sweep gas (N2) flow rate, 1 a.u.; s-lens rf level, 60%; SCAN-mode; resolution, 120000 (at m/z 200); AGC target, 50%; maximum injection time, 200ms.

Quality control samples consisting of pooled samples from each condition were injected at the beginning and periodically through the workflow.

MS/MS acquisition was performed at a resolution of 15000, and the normalized collision energy of the HCD cell was stepped at 10-20-30-40%. This collision energy was tested to obtain appropriate precursor ion and product intensities. The quadrupole isolation window was 1 m/z. The maximum injection time of the C-trap was customized for each inclusion list.

Xcalibur 4.4 software (Thermo Scientific, Waltham, MA, USA) was used for LC/MS instrument control and data processing.

## LC/MS data analysis

Thermo.raw data files were transformed to.mzML using Proteowizard's MSconvert. Then, mzML files were processed using RHermes, a computational tool that improves the selectivity and sensitivity for comprehensive metabolite profiling and identification. RHermes substitutes the conventional untargeted metabolomics workflow that detects and annotates peaks, for an inverse approach that directly interrogates raw LC/MS data points using a comprehensive list of unique molecular formulas that were retrieved from large compound-centric databases (e.g., HMDB, ChEBI, NORMAN). They are used to generate a large set of ionic formulas by combining metabolite molecular formulas with expected adduct ions depending on the polarity.

RHermes solves the limitations of peak detection by finding series of scans, named SOI (Scans Of Interest), which are defined as clusters of data points that match an ionic formula and are concentrated within a short period of time. Identification was performed by HERMES using two strategies: cosine spectral matching using an in-house DB, containing MS/MS spectra from MassBankEU, MoNA, HMBD, Riken and NIST14 databases, and using MassFrontier version 8.0 SR1 (Thermo Scientific, Waltham, MA, USA) matching against the mzCloud database. Spectral hits with high similarity scores (>0.8) were manually revised to assess correct metabolite identifications.

SOIs are qualitative elements that are suited for metabolite annotation and identification. Analytically though, only well-behaved chromatographic peaks are to be quantified, these peaks have a sharp elution profile and a high signal-to-noise ratio. In order to perform SOI quantification, SOI scan series are evaluated and partitioned into well-behaved peaks in order to extract accurate abundance values. This is performed using the qHermes R package that applies the Centwave algorithm originally found in the XCMS R package. Centwave algorithm determines the baseline and boundaries of well-behaved peaks (according to a set of parameters) and assigns an abundance value using the apex (highest value) of the peak, these boundaries may be different from SOI retention time ranges.

After data quality is assured and corrections have been performed, the data is ready for statistical testing.

## Statistical analysis

All the values reported are expressed as median and interquartile range (biochemical and anthropometric variables) or mean and standard deviation (lipidomics data) in accordance with the distribution of the variables. In the same way, differences between groups were calculated using the nonparametric Mann–Whitney test (biochemical and anthropometric variables) or using one-way ANOVA (lipidomics data). P-values <0.05 were statistically significant.

Only SOI ions quantified in >80% of the samples were statistically tested for significant differences across the experimental groups using one-way ANOVA. Statistical results were then adjusted using the False Discovery Rate (FDR) p-value correction method.

Fold-changes, or ratios, are simply calculated by dividing the mean value of an experimental group to a different reference group, for example in case-control studies. They can be used to filter or rank prior statistical results or calculations. In this case, the ratios are calculated using the log2-ratios of the means of the A/B groups. Fold- Changes (FC) larger than 0 indicate a higher intensity in the A group in comparison to the B, and viceversa.

Log2FC and FDR-adjusted p-value thresholds have been tuned to reduce metabolites entities to be identified, otherwise it would not be feasible for manual revision.

Data from significant lipid metabolites concentration expressed in the Log2FC, mean, and standard deviation are given in (S1 and S2 Tables).

## Lipid compound identification

To elucidate the identity of SOI and assign that to features, we have performed a set of fragmentation spectra using LC/MS that physically split the molecular structure into a reproducible and recognizable pattern. These patterns can be compared to those found in reference metabolite databases and matched using similarity analysis. In here, we have used 2 different databases to obtain such similarity: HERMES and MS2ID.

In the case of lipidomics, lipid structures are well-characterized and metabolite spectra can be assigned to a lipid family. This is achieved by the LipidMS R package.

Some databases may simplify lipid naming in a coded manor [28]. Chain notations in phospholipids or glycerolipids may also be noted as X:Y, X being the number of carbons of the fatty acid and Y the number of insaturations.

## Graphical representation

The Hierarchical Clustering Heatmap and the graphical representation of the Pearson Correlation Coefficients between variables were performed using Metabonalyst system [29].

## Results

### Participants

In this study, we carried out a LC/MS-based untargeted lipidomics analysis in serum samples from 209 women. This cohort was made up of 73 normal-weight women and 136 women with morbid obesity. Of these, 65 were metabolically healthy morbid obese and 71 were morbid obese with associated T2DM.

Anthropometrical and clinical parameters of the global studied cohort were evaluated and represented in Table 1. All subjects were comparable in terms of age and sex. Morbid obese

**Table 1. Anthropometric and biochemical data of the study cohort (n = 209): Normal weight vs morbid obesity groups.**

| Variables | Normal Weight (n = 73) Median (25th–75th) | | Morbid Obesity (n = 136) Median (25th–75th) | | p-Value |
|---|---|---|---|---|---|
| Age (years) | 40 | (36–48) | 43 | (36–49) | 0.497 |
| BMI (kg/m$^2$) | 22.58 | (21.31–23.84) | 45.30 | (42.23–49.71) | <0.001 |
| Waist-hip (m) ratio | 0.79 | (0.73–0.85) | 0.92 | (0.86–0.97) | <0.001 |
| Glucose (mg/dL) | 82 | (72–91) | 124 | (114–138) | <0.001 |
| HbA1c (%) | 5 | (4.75–5.25) | 6.1 | (5.32–7.80) | <0.001 |
| Insulin (mUI/L) | 5.93 | (4.54–8.94) | 14.1 | (8.19–23.48) | <0.001 |
| Triglycerides (mg/dL) | 63 | (50.50–85.50) | 129 | (99–158) | <0.001 |
| Cholesterol (mg/dL) | 180 | (162–205.25) | 165.55 | (146.80–188.85) | 0.005 |
| HDL-C (mg/dL) | 65.1 | (54.75–73) | 39.85 | (33.95–46) | <0.001 |
| LDL-C (mg/dL) | 105 | (86.50–123) | 98.50 | (83.12–115.75) | 0.151 |

Data are expressed as median and interquartile range. Significant differences were considered when p-value <0.05. BMI, body mass index; HbA1c, glycosylated hemoglobin A1c; HDL-C, high-density lipoprotein-cholesterol; LDL-C, low-density lipoprotein-cholesterol.

subjects presented significantly higher BMI, waist–hip ratio, glucose, HbA1c, insulin, cholesterol and triglyceride levels, and lower levels of HDL-C than the normal-weight group. Moreover, we did not find significant differences regarding to total cholesterol nor LDL-C levels, which could be explained due to the 28% of subjects with morbid obese were treated with lipid-lowering agents. Additionally, the values of these parameters in the morbid obese cohort are nearly within the reference ranges (total cholesterol reference range: 50–200 mg/dL and LDL-C reference range: <100 mg/dL).

On the other hand, as shown in Table 2, considering only the morbid obese cohort, we found that those patients with associated T2DM presented increased levels of glucose, HbA1c, insulin and triglycerides, and decreased levels of HDL-C that metabolically healthy morbid obese subjects. However, we did not find significant differences in terms of total cholesterol nor LDL-C levels, since as we previously mentioned, the 28% of morbid obese subjects were treated with lipid-lowering agents.

**Table 2. Biochemical variables of the morbid obesity study cohort (n = 136): metabolically healthy morbid obesity group compared to morbid obesity with type 2 diabetes mellitus group.**

| Variables | Metabolically Healthy Morbid Obesity (n = 65) Median (25th–75th) | | Morbid Obesity with Type 2 Diabetes Mellitus (n = 71) Median (25th–75th) | | p-Value |
|---|---|---|---|---|---|
| Age (years) | 41 | (35–49) | 44 | (37–49) | 0.138 |
| BMI (kg/m$^2$) | 44.85 | (41.69–48.14) | 45.39 | (43.41–50.05) | 0.126 |
| Waist-hip (m) ratio | 0.92 | (0.84–0.95) | 0.93 | (0.86–0.98) | 0.053 |
| Glucose (mg/dL) | 87 | (80.50–94.50) | 145.50 | (127.75–190.50) | <0.001 |
| HbA1c (%) | 5.3 | (5–5.65) | 7.2 | (6.30–8.70) | <0.001 |
| Insulin (mUI/L) | 10 | (6.64–16.50) | 19.33 | (13.02–32.48) | <0.001 |
| Triglycerides (mg/dL) | 104 | (82.25–132.25) | 148 | (122–207) | <0.001 |
| Cholesterol (mg/dL) | 164 | (146.37–187.50) | 166.85 | (147.62–190.40) | 0.831 |
| HDL-C (mg/dL) | 42 | (34.95–53.65) | 37.1 | (33–44.50) | 0.023 |
| LDL-C (mg/dL) | 100 | (85.15–119.50) | 95 | (76.40–114) | 0.164 |

Data are expressed as median and interquartile range. Significant differences were considered when p-value <0.05. BMI, body mass index; HbA1c, glycosylated hemoglobin A1c; HDL-C, high-density lipoprotein-cholesterol; LDL-C, low-density lipoprotein-cholesterol.

## Lipid metabolite profile in morbidly obese women in comparison with normal-weight women

First, we have wanted to evaluate the differences of the lipidome between morbid obese group and those normal-weight subjects. In this sense, we have found some lipids that presented significantly differently trends of concentration (Table 3). In this sense, morbid obese subjects presented significantly increased levels of ceramide Cer (42:1;O2), sphingomyelins (SM), diacylglycerols (DG), triacylglycerols (TG), fatty acids and the most of phosphoethanolamines (PE) than the control group. On the other hand, morbid obese subjects had decreased levels of acylcarnitines, bile acids, lyso-phosphatidylcholines (LPC), phosphatidylcholines (PC), the phosphatidylinositol PI (36:2) and PE (O-38:4).

## Lipid metabolite profile in morbid obese women with type 2 diabetes mellitus compared to metabolically healthy morbid obese women

Second, we compared those morbid obese patients with T2DM vs metabolically healthy morbid obese women (Table 4). We reported only increased levels of bile acid deoxycholic acid, DG (36:2), triacylglycerols, phosphatidylcholines, phosphoethanolamines, phosphatidylinositols and the lyso-phosphatidylinositol LPI (16:0). However, we did not find reduced levels of any lipid metabolite in this comparison.

**Table 3. Lipid metabolites concentration increased or decreased in serum samples of morbid obese patients (n = 136) compared to normal-weight subjects (n = 73).**

| Family | | Increased Levels | | | Decreased Levels | |
|---|---|---|---|---|---|---|
| **Acylcarnitines (ACar)** | | | | | ACar (11:0) ACar (9:0) | |
| **Bile acids** | | | | | Deoxycholic acid Glycoursodeoxycholic acid | |
| **Sphingolipids** | **Ceramides (Cer)** | Cer (42:1;O2) | | | | |
| | **Sphingomyelins (SM)** | SM (36:0;O2) SM (36:1;O2) SM (38:4;O2) | SM (42:1;O2) SM (42:2;O3) | | | |
| **Acylglycerols** | **Diacylglycerols (DG)** | DG (36:4) DG (40:7) | | | | |
| | **Triacylglycerols (TG)** | TG (44:3) TG (44:4) TG (45:4) TG (46:4) TG (48:6) TG (52:6;O) | TG (53:6) TG (54:6) TG (54:7;O) TG (54:8) TG (58:3) TG (58:4) | TG (60:11) TG (60:12) TG (60:13) TG (60:4) TG (62:14) | | |
| **Fatty acids** | | Linolenic Acid Hydroxy-docosahexaenoic acid | | | | |
| **Phospholipids** | **Lyso-phosphatidylcholines (LPC)** | | | | LPC (18:0) LPC (18:2) | |
| | **Phosphatidylcholines (PC)** | | | | PC (34:2;O) PC (34:3) PC (35:0) PC (36:4) | PC (38:6;O) PC (38:7) PC (39:5) |
| | **Phosphoethanolamines (PE)** | PE (34:1) PE (36:1) PE (40:5) PE (O-36:2) | | | PE (O-38:4) | |
| | **Phosphatidylinositols (PI)** | | | | PI (36:2) | |

One-way ANOVA test was used to identify significant differences. Data of concentration in log fold change (FC) and p-values, as well as the mean and standard deviation of each lipid specie, are given in (S1 Table).

**Table 4. Lipid metabolites concentration increased in serum samples of morbid obese patients with type 2 diabetes mellitus (n = 71) compared to metabolically healthy morbid obesity subjects (n = 65).** No decreased levels were found.

| Family | | Increased Levels | |
|---|---|---|---|
| Bile acids | | Deoxycholic acid | |
| Acylglycerols | **Diacylglycerols (DG)** | DG (36:2) | |
| | **Triacylglycerols (TG)** | TG (40:1)<br>TG (46:1)<br>TG (46:2)<br>TG (46:4) | TG (48:1)<br>TG (48:2)<br>TG (49:1)<br>TG (54:1) |
| Phospholipids | **Phosphatidylcholines (PC)** | PC (32:1)<br>PC (34:3)<br>PC (35:4) | PC (36:5)<br>PC (36:6)<br>PC (38:3) |
| | **Phosphoethanolamines (PE)** | PE (36:3)<br>PE (38:3)<br>PE (38:4) | PE (40:7)<br>PE (O-38:4) |
| | **Lyso-phosphatidylinositols (LPI)** | LPI (16:0) | |
| | **Phosphatidylinositols (PI)** | PI (32:0)<br>PI (32:1) | |

One-way ANOVA test was used to identify significant differences. Data of concentration in log fold change (FC) and p-values, as well as the mean and standard deviation of each lipid specie, are given in (S2 Table).

## Heatmap of the lipid metabolite profile in women with normal weight, morbid obese women with type 2 diabetes mellitus and metabolically healthy morbid obese women

Finally, we wanted to perform a hierarchical clustering analysis using a heatmap to visualize which lipid metabolites previously identified present a characteristic profile across the studied groups (Fig 1). In this regard, we observe a set of lipid metabolites that clearly exhibit higher levels in subjects with normal weight (class 1) compared to all morbid obese patients (classes 2 and 3). These lipids align in the majority of cases with significantly decreased metabolites in patients with morbid obese compared to the control group (Table 3). On the other hand, we note another set of lipid metabolites that show significantly elevated levels in those morbid obese subjects with associated T2DM (class 3) compared to controls (class 1) or metabolically healthy morbid obese individuals (class 2). Finally, we identify a large group of lipid metabolites that display lower levels in normal-weight subjects (class 1) compared to morbid obese subjects (classes 2 and 3); notably, some of these metabolites are significantly more elevated in metabolically healthy morbid obese women (class 2), while another subgroup shows more elevated levels in women with T2DM associated to morbid obese (class 3). Several of the characteristic metabolites of the class 3 profile or group of women with morbid obese and associated T2DM coincide with those found increased in Table 4.

## Correlation analysis

To complete our lipidomics study, we have performed a correlation analysis to evaluate the association between the lipid metabolites and anthropometric and biochemical parameters of our study cohort of women. These results were reported in (S3 Table). From this analysis, we can synthesize that acyl carnitines tend to show negative associations with BMI, waist-to-hip ratio, glucose levels, hbA1c and triglycerides, while showing positive associations with cholesterol and HDL-C. On the other hand, ceramides, sphingomyelins, glycerolipids (DG and TG) and linoleic acid predominantly show positive associations with BMI, waist-to-hip ratio, glucose levels, insulin, glycated hemoglobin and triglycerides, while showing negative associations

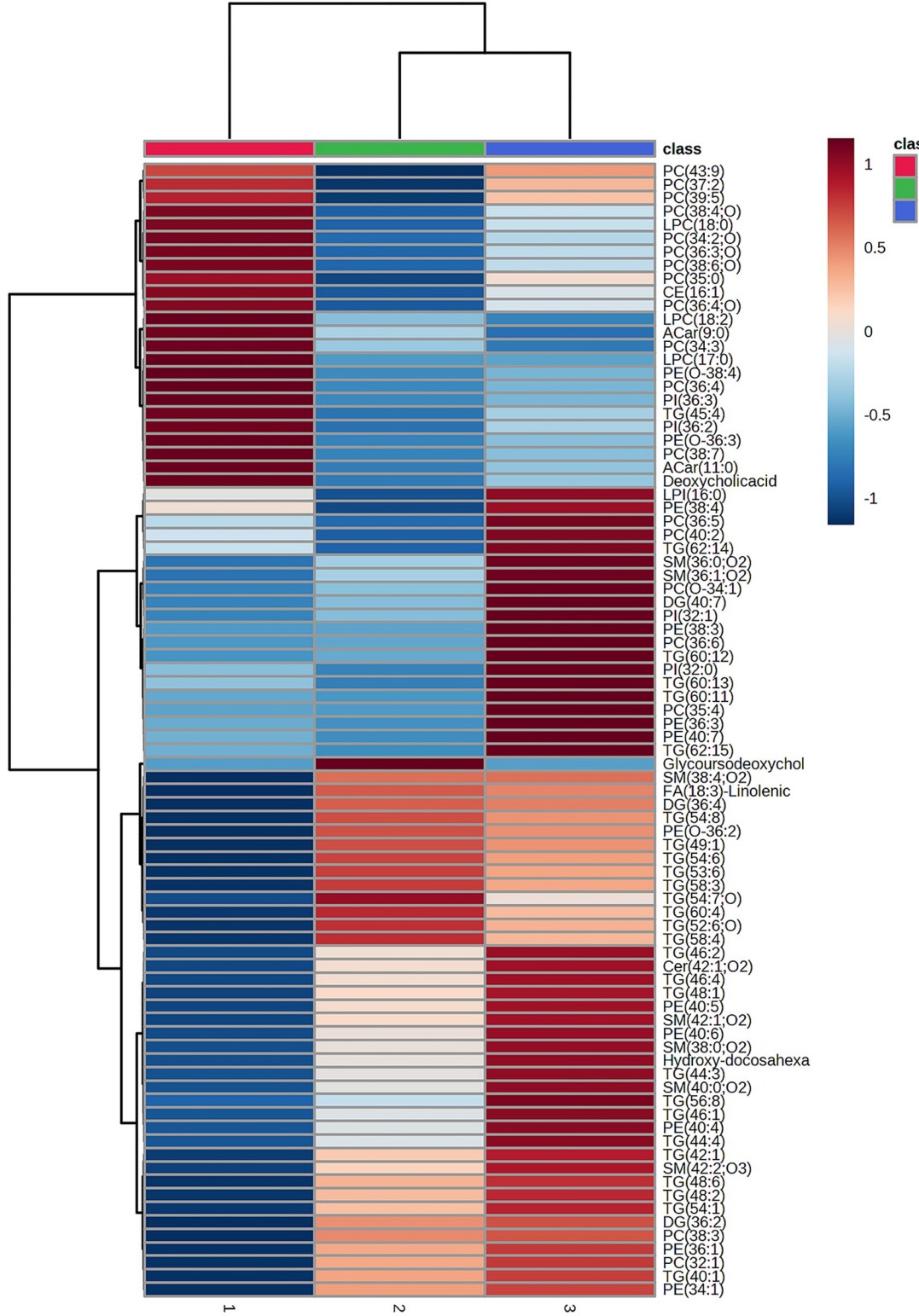

**Fig 1. Hierarchical clustering heatmap of the lipid metabolites identified across the studied groups.** The groups included in the analysis were as follows: 1) In red color normal-weight women (n = 73); 2) In green color metabolically healthy morbid obese women (n = 65); 3) In blue color women with morbid obesity and type 2 diabetes mellitus (n = 71). The graphic show only group average of every metabolite value. Ward clustering method with Euclidean distance measure were used in this analysis. The intensity range is from +1 (red) to -1 (blue) representing most increased or most decreased levels of metabolites. This graph

was performed using Metabonalyst system. Phosphatidylcholine (PC), lyso-phosphatidylcholine (LPC), cholesteryl-ester (CE), acyl-carnitine (ACar), lyso-phosphatidylinositol (LPI), phosphoethanolamine (PE), phosphatidylinositol (PI), triacylglycerol (TG), sphingomyelin (SM), diacylglycerol (DG), fatty acid (FA), ceramide (Cer).

with HDL-C levels. In the case of phospholipids, there is less homogeneity in the correlations. We can deduce a tendency of positive correlations between phosphatidylcholines and phosphatidylinositols levels and cholesterol, HDL-C and LDL-C levels. While they show some negative correlations with parameters of sugar metabolism and BMI, except for PC(38:3), PC(O-34:1) and some phosphoethanolamines.

## Discussion

This study lies in the fact that we have identified a panel of lipid metabolites in serum samples of women with morbid obesity. Moreover, we have identified some lipid metabolites specifics from women with T2DM associated to morbid obese that permits to discriminate this group from a cohort of metabolically healthy morbid obese women.

In the context of obesity and T2DM, previous studies employing untargeted lipidomics in prospective cohorts have underscored the pivotal role of such investigations in biomarker development, understanding pathophysiology, personalized medicine tool development, therapeutic target identification, early disease detection, and population risk stratification [30–32]. These studies have illuminated the intricate lipidomic alterations associated with these metabolic conditions, offering invaluable insights into their etiology and progression.

### Lipidome comparative between morbidly obese and normal-weight women

In this study of lipidomics, we have found increased levels of ceramides (Cer), sphingomyelins (SM), diacyl and triacylglycerols (DG and TG), fatty acids and the most of phosphoethanolamines (PE) in the morbid obese cohort of women compared to normal-weight women. We also have found decreased levels of acylcarnitines, bile acids, lyso-phosphatidylcholines (LPC), phosphatidylcholines (PC), phosphatidylinositols (PI) and PE (O-38:4). In this sense, given that we previously performed an untargeted metabolomics study in the same cohort of subjects, we agree with our previous results in terms of SM (36:1;O2), DG (36:4), TG (54:6), PE (34:1) and linoleic acid [22], which were found to be increased in the morbid obese cohort in both analyses. On the other hand, this current lipidomics study also agree with our previous metabolomics study regarding to deoxycholic and glycoursodeoxycholic acids, LPC (18:0) and LPC (18:2), PC (34:2;O) and PC (34:3) and PI (36:2), which were found to be decreased in obese cohort in both studies. Conversely, in this lipidomics study we have found increased levels of PE (40:5), while we found decreased levels of this specie of metabolite in morbid obese cohort in the metabolomics study. Additionally, we have found decreased levels of PC (36:4) in the morbid obese cohort in this lipidomics study, but we found it increased in the metabolomics assay [22].

### Lipidome comparative between morbidly obese with type 2 diabetes mellitus and metabolically healthy morbid obese women

Then, we compare the findings of this lipidomics study and our previous metabolomics analysis concerning on the metabolites significantly increased or decreased in the cohort of morbid obese women with T2DM vs metabolically healthy morbid obese women. In this study, we reported increased levels of deoxycholic acid, DG (36:2), triacylglycerols, phosphatidylcholines, phosphoethanolamines, phosphatidylinositols and lyso-phosphatidylinositol LPI (16:0).

Compared to the previous metabolomic study, we only found increased levels of PC (36:5) in both studies, meanwhile we found increased levels of PC (32:1) and PC (34:3) in the lipidomics study but we found them decreased in the metabolomics assay [22].

Some of these discrepancies in both comparisons can be explained given that although the same samples were analysed in both analyses, these studies were performed in different times, and the accuracy of the identification of the metabolite/lipid could differ in some cases [33]. Moreover, metabolomics and lipidomics analyses can give disparate results because are techniques that analyse different fractions of a biological sample (polar or hydrophobic phases), and although the results may coincide and therefore be validated, they may differ [33].

Concerning the existing literature, carnitines play a key role transporting fatty acids through the cell for its oxidation, so the accumulation of acylcarnitines may suggest a deregulation in the fatty acid oxidation process [34]. In this sense, increased levels of circulating acylcarnitines have been found in metabolomics studies in patients with obesity and/or T2DM [35,36]. However, it was reported that some dietary components and changes in the diet can modulate circulating acylcarnitines profile [34,37]. In this regard and given that our morbid obese subjects underwent a very-low calorie diet before the surgical intervention, it makes sense that we found decreased levels of some acylcarnitines compared to normal-weight subjects who followed a regular diet prior the blood sample obtention.

On the other hand, we found decreased levels of bile acids in our morbid obese patients compared to normal-weight subjects. In this context, bile acids regulated the intestinal absorption of lipids and also modulate the lipid metabolism in other tissues by activating farnesoid X receptor and G-protein-coupled bile acid receptor-1 [38]. Therefore, it makes sense that bile acid metabolism could be altered in obesity and T2DM conditions [39].

In terms of ceramides, we found increased levels in our morbid obese subjects. Ceramides are lipid molecules that serve as the structural unit of complex sphingolipids. They can stimulate fatty acid uptake in cells and the *de novo* lipogenesis in liver. Moreover, ceramides also inhibit lipolysis and brake-down insulin signalling [40]. The production of ceramides used to be induced in response to stress stimuli such as nutritional overload, inflammation, oxidative stress, and hypoxia [41]. High circulating levels of ceramides have been found as associated factor to obesity, T2DM and other metabolic disorders [42,43]. In addition, in a metabolomics study, also found a significant association of BMI with Cer (42:1) in women [44], just like in our study.

Sphingomyelins are the most abundant circulating sphingolipids [45], that were reported to be increased in obese subjects and being correlated with triacylglycerol and LDL-C levels and insulin resistance parameters [46,47]. In addition, in a metabolomics study in Korean men with obesity, they also found increased levels of sphingomyelins [48]. In this case, Beyene *et al.* agrees our results in women concerning sphingomyelins, since they also found a significant positive association between BMI and SM (36:1) and SM (42:1) levels [18].

In terms of diacyl and triacylglycerol, we found increased levels of these lipids in our morbid obese women, which was reported that their increase are usually linked glucose and lipid metabolism disruption [49]. Another study of Beyene *et al.*, also reported a significant positive association of BMI with DG (36:4), TG (54:6) and TG (54:7) in women [44].

Concerning linoleic acid, that is an omega-6 poly-unsaturated fatty acid, it may also present metabolic disturbing effects [50]. It was previously reported that this fatty acid can induce weight gain and stimulate the release of influencing-mediators of the metabolism and pro-inflammatory factors [51]. For this reason, it makes sense that our women with morbid obese presented increased levels of linoleic acid. On the other hand, we reported increased levels of the hydroxy-docosahexaenoic acid in obesity. This fatty acid is an oxylipin that has reported to induce apoptosis and oxidative stress [52].

Regarding phospholipids, that are the major components of biological membranes [53], we found reduced levels of lyso-phosphatidylcholines in obesity. In this context, some previous metabolomics reports also found decreased levels of lyso-phosphatidylcholines in obesity [54,55], specifically the species LPC (18:0) and LPC (18:2) in women [44], and the specie LPC (18:2) again in cohorts with men and women [56–58]. In this regard, in a recent review, authors stated that given that obesity is associated with a state of low-grade chronic inflammation, phosphatidylcholines and lyso-phosphatidylcholines are becoming important in the study of obesity. However, the metabolomic results used to be controversial in terms of these phospholipids. In any case, the most common trend is a negative association between BMI and most of phosphatidylcholines and lyso-phosphatidylcholines [53]. Again, Beyene *et al.* found a negative significant association between phosphatidylcholines PC (34:2), PC (36:4) and BMI in women in two different metabolomic studies [18,44], which agree our results. However, they also found increased levels of PC (39:5), PC (34:3) and PC (38:7) [18,44], while we have found decreased levels of these metabolites.

Deregulation of phosphoethanolamine metabolism was reported to induce metabolic disorders such as insulin resistance and obesity; however, the precise mechanism is not clearly understood [59]. Regarding, phosphatidylinositol, low levels of this phospholipid were reported to be related to obesity-induced inflammation and insulin resistance [19]. In our study, we found the most of phosphoethanolamines increased except for PE (O-38:4), and we found decreased levels of PI (36:2).

## Lipidome of both comparatives

Then, when we compared the lipidomic profile between morbid obese women with T2DM and metabolically healthy morbid obese, we found increased levels of deoxycholic acid in the T2DM subjects. In this sense, preclinical studies suggested that insulin resistance and hyperglycaemia induce the bile acids production, triggering an alteration in the bile acids homeostasis [60]. Moreover, the presence of T2DM has been related to increased circulating levels of some bile acids such as deoxycholic acid [61–63].

On the other hand, we found higher levels of DG (36:2) in our T2DM women. Previous reports identified that the increase of diacylglycerol levels in plasma are induced by hyperglycaemia [64]. Also, we found increased levels of some species of triacylglycerols in our T2DM subjects, which is reinforced by literature, that stated elevated triglyceride levels are common in patients with T2DM [65], since these increased levels were linked to insulin resistance [66].

In this group of women with morbid obese and T2DM, we presented increased levels of phospholipids, such as phosphatidylcholines, phosphoethanolamines, LPI (16:0), PI (32:0) and PI (32:1). In this sense, recent research have focused mainly on the relationship between insulin sensitivity and phospholipids, suggesting that phospholipid composition of cell membranes may modulate the action of insulin [67]. Additionally, lyso-phosphatidylinositols, which are bioactive lipids generated by lipases, seems to play a key role in several physiological and pathological processes [68]. Thus, altered levels of lyso-phosphatidylinositol have been found in obesity and T2DM [69]. In this context, in a previous lipidomic study in Finnish men, authors found increased levels of PC (32:1), PC (34:2e) and PC (36:1) in subjects with metabolic risk factors and a high probability to progress into T2DM [70]. Other lipidomic study reported increased levels of phospholipids in patients with T2DM [20]. However, other untargeted lipidomic works suggested an inverse association between concretely PC (34:3) and lyso-phosphatidylcholine levels with T2DM [58,71]. In another study, authors reported a positive association between triacylglycerols and PC (34:3) levels with insulin resistance, the same than in our study, while these authors also informed from a negative association between PE (38–4)

levels and insulin resistance [72], contrasting our results that reported only increased levels of this lipid metabolite.

Finally, taking into account the two comparatives from this lipidomics study, we reported similarities such as increased levels of diacylglycerols and TG (46:4) in morbid obese women compared to normal-weight women, and also in morbid obese women with T2DM compared to metabolically healthy morbid obese. Conversely, we reported decreased levels of deoxycholic acid, PC (34:3) and PE (O-38:4) in our morbid obese women compared to normal-weight subjects, but we found the levels of these lipid metabolites being increased in our morbid obese women with T2DM compared to metabolically healthy morbid obese women. In this sense, these metabolites with opposite trends should be further investigated as potential factors of metabolic risk of T2DM in morbid obese women. The results of this comparative could be the most interesting results of this study both of pathophysiological mechanisms and of possible markers of obesity with diabetes.

## Limitations

The possible limitation of this study are that although we have conducted the analysis in a homogeneous cohort of morbid obese women, it is difficult to compare these results with the literature due to the differences between the cohorts, since it can generate a bias. On the other hand, we could not control some parameters such as renal function, which could affect lipid profile and metabolic homeostasis. Moreover, the 28% of subjects with morbid obese were treated with lipid-lowering agents, which could affect the obtained lipidome. In addition, given that these results seem promising, it is necessary to perform targeted omics analyses in larger validation groups to confirm these findings. Additionally, LC/MS has some technical conditions: LC tools are sensitive to temperature and atmospheric pressure changes, solvent impurities or hardware inconsistencies. MS might detune during an experiment or lose sensitivity during an experiment. However, in this untargeted lipidomics analysis, quality control samples were used in order to evaluate any issues, in addition to human error. These contain a pool of all or a subset of the analyzed biological samples, and therefore they are equivalent in terms of matrix and lipid profile composition. Quality control samples were injected regularly between biological samples during the experiment.

## Conclusion

In this LC/MS-based untargeted lipidomics study, meticulously conducted on a cohort of women with MO, our investigation has reported a lipid metabolite profile associated with obesity. Also, a specific lipid profile emerged when comparing morbid obese women with T2DM to metabolically healthy morbid obese counterparts. Notably, distinctive trends in the concentrations of deoxycholic acid, PC (34:3), and PE (O-38:4) were observed between these comparative groups. These metabolites with opposite trends should be further investigated as potential factors of metabolic risk of T2DM in morbid obese women. Also, these insights can help us by advancing our research in physiopathology of metabolic alterations associated with obesity and T2DM, with the purpose to develop or targeted diagnostic and individualized therapeutic interventions.

## Supporting information

**S1 Table. Log fold change (FC), mean and standard deviation (SD) of each lipid specie from the morbid obese (MO) and the normal-weight (NW) groups, and p-values.** Data from significant lipid metabolites concentration expressed in the Log2FC, mean, and standard deviation. One-way ANOVA test was used to identify significant differences (adjusted p-value

<0.05 was considered significant).
(XLSX)

**S2 Table. Log fold change (FC), mean and standard deviation (SD) of each lipid specie from the morbid obese with type 2 diabetes mellitus groups (MO&T2DM) and the metabolically healthy morbid obese (MHMO) groups, and p-values.** Data from significant lipid metabolites concentration expressed in the Log2FC, mean, and standard deviation. One-way ANOVA test was used to identify significant differences (adjusted p-value <0.05 was considered significant).
(XLSX)

**S3 Table. Correlation analysis between lipid metabolites and anthropometric and biochemical parameters.** Bivariate Spearman correlation test was used to identify associations. Rho (Spearman correlation coefficient) indicates the strength of association and p-value indicates the significance (p-value<0.05 was considered significant). Green color was used for positive correlations and orange color was used for negative correlations.
(XLSX)

## Acknowledgments

Thanks to the Universitat Rovira i Virgili for its administrative collaboration.

## Author Contributions

**Conceptualization:** Cristóbal Richart.

**Data curation:** Laia Bertran, Jordi Capellades.

**Formal analysis:** Laia Bertran, Cristóbal Richart.

**Funding acquisition:** Cristóbal Richart.

**Investigation:** Laia Bertran, Cristóbal Richart.

**Methodology:** Jordi Capellades, Sonia Abelló, Carmen Aguilar.

**Project administration:** Cristóbal Richart.

**Resources:** Sonia Abelló, Carmen Aguilar, Teresa Auguet.

**Software:** Laia Bertran, Jordi Capellades.

**Supervision:** Cristóbal Richart.

**Validation:** Teresa Auguet, Cristóbal Richart.

**Visualization:** Cristóbal Richart.

**Writing – original draft:** Laia Bertran.

**Writing – review & editing:** Cristóbal Richart.

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
