## [Decision Letter · Decision Letter 0]

1 Apr 2024

PONE-D-24-07665Untargeted lipidomics analysis in women with morbid obesity and type 2 diabetes mellitus: a comprehensive studyPLOS ONE

Dear Dr. Richart,

Thank you for submitting your manuscript to PLOS ONE. After careful consideration, the reviewers have identified minor concerns that require addressing prior to possible acceptance. Please address within the text of the manuscript as noted in their reviews.

We look forward to receiving your revised manuscript.

Kind regards,

Jon M. Jacobs, Ph.D.

Academic Editor

PLOS ONE

Journal Requirements:

"Thanks to the Universitat Rovira i Virgili for its administrative collaboration and Fundació URV for its collaboration in the funding of this study (Project IT20041-S to C.R)."

"The authors received no specific funding for this work."

Reviewers' comments:

Reviewer's Responses to Questions

**Comments to the Author**

1. Is the manuscript technically sound, and do the data support the conclusions?

Reviewer #1: Yes

Reviewer #2: Yes

2. Has the statistical analysis been performed appropriately and rigorously? 

Reviewer #1: No

Reviewer #2: Yes

3. Have the authors made all data underlying the findings in their manuscript fully available?

Reviewer #1: Yes

Reviewer #2: Yes

4. Is the manuscript presented in an intelligible fashion and written in standard English?

Reviewer #1: No

Reviewer #2: Yes

5. Review Comments to the Author

Reviewer #1: • In the methodology section authors say that “The study cohort of this study was composed by only women to evaluate a homogenous cohort of subjects to avoid the interference of confounding factors such as sex”. This is a valid point. However, authors need to consider several other confounding factors, mainly statin users between the groups. Other factors that could affect serum lipid levels include alcohol intake and smoking. Also, there is no information on whether these patients are hypertensive.

• Correlation analysis between key parameters could better help to identify the linking of specific lipids with other parameters.

• Also, an additional table providing Anthropometric and biochemical data of the study subgroup one with T2DM (71) and the (n= 65) with metabolically healthy morbid obese cases. This table could essentially provide more insights.

• Article misses discussion from the important information on untargeted lipidomics data from prospective studies (For example: PMID: 35035379)

Reviewer #2: The article sent for evaluation is very interesting and fits into current scientific trends.

The introduction is a very good introduction to the topic.

Methodology - patients were correctly classified following the adopted assumptions.

I was only wondering about the issue of menopause, which significantly affects metabolism. On the other hand, the average age in both groups is similar; hence, the share of women at menopausal age should be comparable. However, this does not exclude early menopause at the age of, for example, 40. If researchers do not have such information, it should be included in the limitation of the study section. The laboratory data lacked information on kidney function assessment, which significantly affects the body's lipid metabolism. I also found no information based on recruitment for the study, especially for women with normal body weight. What about comorbid conditions apart from diabetes mellitus?

The discussion is interesting, but you can get lost, so I suggest the authors divide it into small sections.

There are no applications; they need to be supplemented.

I recommend adding a limitation to the study section.

It would be helpful to have a diagram of the research, which was done one by one, so that the reader could sort it all out.

A graphical abstract would also be helpful - where the authors would present their most important discovery.

The two schemes I mentioned could be combined into one - it's up to the researchers to decide.

6. PLOS authors have the option to publish the peer review history of their article (what does this mean?). If published, this will include your full peer review and any attached files.

Reviewer #1: No

Reviewer #2: No

---

## [Author Response · Author response to Decision Letter 0]

4 Apr 2024

Reviewer #1: 

• In the methodology section authors say that “The study cohort of this study was composed by only women to evaluate a homogenous cohort of subjects to avoid the interference of confounding factors such as sex”. This is a valid point. However, authors need to consider several other confounding factors, mainly statin users between the groups. Other factors that could affect serum lipid levels include alcohol intake and smoking. Also, there is no information on whether these patients are hypertensive.

Answer: First of all, thank you for your valuable feedback, which has greatly contributed to the refinement of our manuscript.

We thoroughly understand your comment and wholeheartedly agree with your observation. Our cohort of women in this study is derived from a collection of patients/samples, predominantly comprising women, with fairly specific inclusion and exclusion criteria. It is indeed true that we avoided including patients with an alcohol consumption exceeding 10g per day, as well as those with a history of recurrent tobacco use, as these factors can evidently impact metabolic parameters. Upon verifying these data, we confirm that all patients meet these criteria. Therefore, in the Materials and Methods section, under exclusion criteria, we have added the following phrase (lines 112-114, pages 5-6): “In addition, women with an alcohol intake exceeding 10g per day and recurrent smokers have been excluded from this study.”

Furthermore, regarding the use of statins, we were unable to verify this information as we only have access to whether patients are taking lipid-lowering agents (not statins or fibrates). As discussed in the Results section (line 241, page 11), 28% of our cohort (comprising morbidly obese individuals) are taking this type of medication. Consequently, we have included this information again in the Limitations section (line 505-507, page 24): “Moreover, the 28% of subjects with morbid obese were treated with lipid-lowering agents, which could affect the obtained lipidome.”

Lastly, we would like to note that we have not included blood pressure data in Tables 1 and 2. Although 39.8% of the patients suffer from hypertension, 98% of them are taking medication for high blood pressure. Therefore, no differences in blood pressure were observed among the different study groups.

• Correlation analysis between key parameters could better help to identify the linking of specific lipids with other parameters.

Answer: We agree with your suggestion and have added a section at the end of the results section with the correlation analysis of all lipid metabolites with various biochemical and anthropometric parameters of our study cohort (lines 327-341, page 17). Although these correlations have been discussed in the text, we have added a supplementary table (S3 Table) with all the correlations.

• Also, an additional table providing Anthropometric and biochemical data of the study subgroup one with T2DM (71) and the (n= 65) with metabolically healthy morbid obese cases. This table could essentially provide more insights.

Answer: In this regard, we already have included this table (Table 2). However, it is true that this table did not include anthropometric data such as age, BMI and waist-hip ratio. In this sense, we have included this information in the new Table 2 (page 12).

• Article misses discussion from the important information on untargeted lipidomics data from prospective studies (For example: PMID: 35035379)

Answer: Dear reviewer, we agree with your suggestion and have added a paragraph in the discussion section (lines 347-353, page 18) where this article you suggested, along with two other related ones, is cited, defining the importance and applicability of these untargeted lipidomics studies in prospective cohorts: “In the context of obesity and T2DM, previous studies employing untargeted lipidomics in prospective cohorts have underscored the pivotal role of such investigations in biomarker development, understanding pathophysiology, personalized medicine tool development, therapeutic target identification, early disease detection, and population risk stratification [30–32]. These studies have illuminated the intricate lipidomic alterations associated with these metabolic conditions, offering invaluable insights into their etiology and progression.”

Reviewer #2: 

- The article sent for evaluation is very interesting and fits into current scientific trends. The introduction is a very good introduction to the topic. Methodology - patients were correctly classified following the adopted assumptions. I was only wondering about the issue of menopause, which significantly affects metabolism. On the other hand, the average age in both groups is similar; hence, the share of women at menopausal age should be comparable. However, this does not exclude early menopause at the age of, for example, 40. If researchers do not have such information, it should be included in the limitation of the study section. The laboratory data lacked information on kidney function assessment, which significantly affects the body's lipid metabolism. I also found no information based on recruitment for the study, especially for women with normal body weight. What about comorbid conditions apart from diabetes mellitus?

Answer: First, thank you for your experienced feedback, which has greatly contributed to the improvement of our manuscript. Regarding your concern about menopause, I would like to point out that we did not recruit menopausal women or those with severe premenopausal symptoms, nor did we include patients using hormonal contraceptives or similar treatments. This was done to avoid hormonal and metabolic biases in our results, and this information was already detailed in the exclusion criteria of our study (lines 111-112, page 5).

As for the lack of information on the renal function of the patients, we acknowledge that this is an important limitation that should be mentioned in its corresponding section (lines 504-505, pages 24).

Lastly, I would like to highlight that although we have focused on diabetes mellitus due to its impact on the lipid profile of our patients, our cohort mainly consists of women with morbid obesity, many of whom suffer from other comorbidities such as dyslipidemia and hypertension. Regarding dyslipidemia, as detailed in limitations (lines 505-507, page 24), approximately 28% of the patients are under treatment for this condition, which practically represents all those with dyslipidemia in our obesity cohort (which constitutes 30% of the obese cohort).

Additionally, as mentioned earlier, approximately 39.8% of obese women suffer from hypertension, but the vast majority of them are under treatment. Therefore, it is possible that these diseases are "nullified" in terms of biochemical differences between the groups, as they remain within normal ranges, as described in the results (line 242-244, page 11).

Moreover, it's important to mention that both normal-weight patients and those with metabolically healthy morbid obesity do not suffer from any metabolic alterations.

Regarding the recruitment of patients, although we have defined that women with morbid obesity are part of patients undergoing bariatric surgery and have established the sample collection period and access to clinical data for the entire cohort, we have now added that women with normal weight are volunteers who meet the criteria for metabolic health (normal BMI and no defined metabolic alterations) (lines 97-98, page 5).

- The discussion is interesting, but you can get lost, so I suggest the authors divide it into small sections.

Answer: I appreciate your comment and we have subdivided the Discussion section to make it more readable into: 

• Lipidome comparative between morbidly obese and normal-weight women

• Lipidome comparative between morbidly obese with type 2 diabetes mellitus and metabolically healthy morbid obese women

• Lipidome of both comparatives 

• Limitations

- There are no applications; they need to be supplemented. I recommend adding a limitation to the study section. It would be helpful to have a diagram of the research, which was done one by one, so that the reader could sort it all out. A graphical abstract would also be helpful - where the authors would present their most important discovery. The two schemes I mentioned could be combined into one - it's up to the researchers to decide.

Answer: According to your suggestion, we have uploaded a graphical abstract/diagram of our study to make them easier to understand (file name: Graphical_abstract.tif).

---

## [Decision Letter · Decision Letter 1]

29 Apr 2024

Untargeted lipidomics analysis in women with morbid obesity and type 2 diabetes mellitus: a comprehensive study

PONE-D-24-07665R1

Dear Dr. Richart,

We’re pleased to inform you that your manuscript has been judged scientifically suitable for publication and will be formally accepted for publication once it meets all outstanding technical requirements.

Kind regards,

Jon M. Jacobs, Ph.D.

Academic Editor

PLOS ONE

Additional Editor Comments (optional):

Reviewers' comments:

Reviewer's Responses to Questions

**Comments to the Author**

1. If the authors have adequately addressed your comments raised in a previous round of review and you feel that this manuscript is now acceptable for publication, you may indicate that here to bypass the “Comments to the Author” section, enter your conflict of interest statement in the “Confidential to Editor” section, and submit your "Accept" recommendation.

Reviewer #1: (No Response)

Reviewer #2: All comments have been addressed

2. Is the manuscript technically sound, and do the data support the conclusions?

Reviewer #1: Yes

Reviewer #2: Yes

3. Has the statistical analysis been performed appropriately and rigorously? 

Reviewer #1: Yes

Reviewer #2: Yes

4. Have the authors made all data underlying the findings in their manuscript fully available?

Reviewer #1: Yes

Reviewer #2: Yes

5. Is the manuscript presented in an intelligible fashion and written in standard English?

Reviewer #1: Yes

Reviewer #2: Yes

6. Review Comments to the Author

Reviewer #1: Authors have answered all my queries and I donot have any additional suggestions for this manuscript

Reviewer #2: The authors have addressed all my comments. I think the article is ready to be accepted and published.

7. PLOS authors have the option to publish the peer review history of their article (what does this mean?). If published, this will include your full peer review and any attached files.

Reviewer #1: No

Reviewer #2: No

---

## [Editor Report · Acceptance letter]

2 May 2024

PONE-D-24-07665R1 

PLOS ONE

Dear Dr. Richart, 

I'm pleased to inform you that your manuscript has been deemed suitable for publication in PLOS ONE. Congratulations! Your manuscript is now being handed over to our production team.

Kind regards, 

on behalf of

Dr Jon M. Jacobs 

Academic Editor

PLOS ONE